# A Review of Research on Embodied Carbon in International Trade

**Haoran Wang * and Toshiyuki Fujita**

Graduate School of Economics, Kyushu University, Fukuoka 819-0395, Japan;
fujita.toshiyuki.865@m.kyushu-u.ac.jp
* Correspondence: wang.haoran.714@s.kyushu-u.ac.jp

**Abstract:** Nowadays, how to reduce carbon emissions is a hot issue in environmental economics research, and countries around the world are having extensive discussions on their respective carbon emission obligations. The embodied carbon contained in international trade plays a crucial role in controlling pollutant emissions but it is often overlooked, resulting in problems such as carbon displacement and avoidance of responsibility for pollutant emissions. Based on the Social Sciences Citation Index (SSCI) and Science Citation Index-Expanded (SCI-E) database, this paper adopts a bibliometric method to summarize 626 papers from 1994 to 2023 in six aspects, including the number of the literature, the literature citations, research region, journal, author, and research discipline. Meanwhile, the research method and model used in the collected papers are classified and reviewed. Then, this study briefly outlines the current status of embodied carbon emissions and the international pollutant identification laws and analyzes the shortcomings of existing research and the rationality of responsibility identification principles. Finally, we propose future research hotspots by combining carbon neutrality and carbon trading theory.

**Keywords:** embodied carbon; international trade; input–output model; environment; bibliometric method

## 1. Introduction

With the continuous improvement of the industrial level and the gradual expansion of human activities in countries around the world, environmental problems are becoming more and more serious and gradually becoming the biggest obstacle to social progress. Over the past half-century, the earth's ecology and climate have undergone tremendous changes. Excessive greenhouse gas emissions have directly contributed to global warming and caused problems related to flora, fauna, water, and energy. According to the Fifth Assessment Report (2013) of the Intergovernmental Panel on Climate Change (IPCC), global $CO_2$ concentrations have increased by 40% since industrialization, and oceans have absorbed 30% of anthropogenic $CO_2$ emissions, which directly contributes to ocean acidification [1]. The Sixth Assessment Report (2022) states that approximately 17% of historical cumulative net $CO_2$ emissions since 1850 occurred between 2010 and 2019 and global $CO_2$ emissions are rising every year [2]. In addition, global $CO_2$ emissions reached a record high of 36.3 gigatons (Gt) in 2021 from the International Energy Agency Data. Carbon emission reduction has become a crucial issue that countries around the world must face.

In 1992, the world's first international convention for comprehensive control of greenhouse gas emissions, the United Nations Framework Convention on Climate Change (UNFCCC), was signed in Brazil. Subsequently, countries signed the Kyoto Protocol in 1997, making the reduction of greenhouse gas emissions an obligation of developed countries. In 2015, the Paris Agreement was signed to replace the marginally effective Kyoto Protocol, making it clear that the UNFCCC will be strengthened to strictly control global climate change. In the context of nowadays global integration, international trade will

have a significant impact on the use of natural resources and environmental changes, and directly affect the emission of greenhouse gases and pollutants. The most important issues are the transfer of embodied carbon and its emission.

The purpose of this paper is to analyze the development and trends of research on embodied carbon in international trade in the past 30 years by using bibliometric methods. Firstly, we use the Web of Science database to collect the relevant literature on embodied carbon research, summarize and review the number, citations, hotspots regions, authors, journals, and research discipline. Secondly, we summarize the main research methods. Thirdly, we briefly discuss the current research status of international trade embodied carbon and carbon emissions, including the impact of trade on carbon emissions and carbon flows and the law of pollutant identification. Finally, combined with the carbon trading market and carbon neutralization, we discuss the future development trend of embodied carbon and draw a conclusion.

## 2. Overview of the Literature on Embodied Carbon Research

Along with the expansion of global trade and the gradual change of the climate, the issue of carbon emission and embodied carbon caused by trade development has gradually become a research hotspot. The concept of embodied carbon can be traced back to 1974 when the "embodied flow" was first introduced by the International Federation of Institutes of Advanced Study (IFIAS). They believe that embodied with the name of each resource is that which can be used to express or calculate the total amount of direct and indirect consumption of a resource during its production or service, such as in relation to embodied energy, embodied water, etc. [3]. Embodied carbon is derived from the embodied flow, which refers to the direct or indirect carbon emissions generated by the consumption of carbon-containing raw materials in the production process of products. The embodied carbon in international trade is the carbon emissions generated in the process of international trade exchange. Since the 1990s, studies related to embodied carbon have received extensive attention from scholars worldwide and gradually become a research hotspot. In this paper, we selected Social Science Citation Index (SSCI) and Science Citation Index Expanded (SCI-E), the sub-databases of Web of Science Core Collection, as the source of the literature data. The literature data sources were searched for with the keywords "carbon emission", "international trade", and "embodied carbon", and "article" or "review article" was selected as the type of literature, and 627 search results were obtained (search date 15 February 2023). Excluding one of the literature with low relevance, a total of 626 valid papers were obtained.

### 2.1. The Literature Statistics

The first article on embodied carbon in international trade was published in 1994 by Wyckoff AW and Roop JM in Energy Policy, which examined the embodied carbon involved in trade imports for six OECD countries [4], but the environmental and pollutant emission issues arising from trade at that time did not attract widespread attention. As seen in Figure 1, there was a paucity of research on embodied carbon until 2005, and then it began to pick up in 2006. In 2007, the Fourth Assessment Report brought attention to the issue of carbon emissions and research began to grow. With the publication of the fifth IPCC report in 2014 and the signing of the Paris Agreement in 2015, people have realized the seriousness of such problems and the urgency of emission reduction; thus, related research began to spurt and reached its peak around 2018 and gradually remained stable.

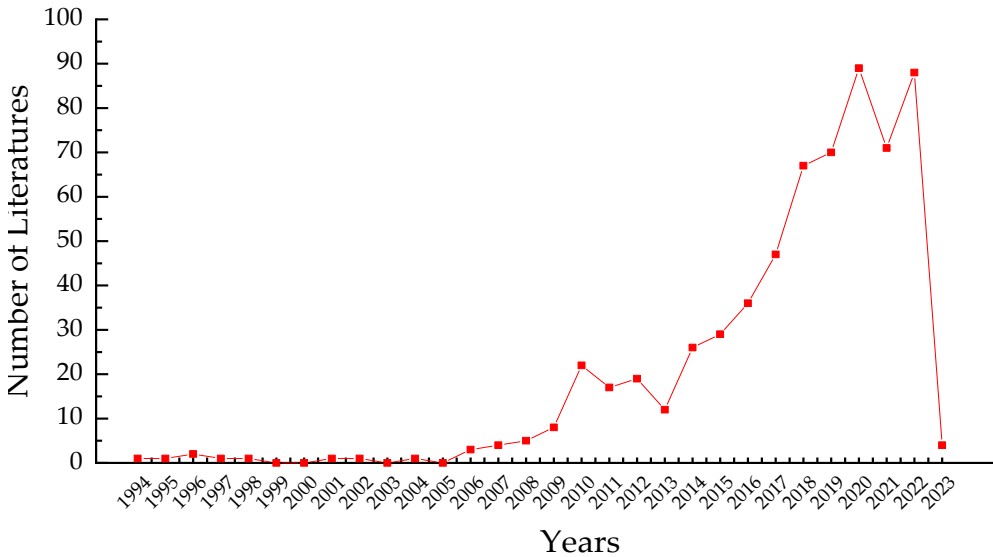

**Figure 1.** The number of the annual literature on embodied carbon.

*2.2. Citation Statistics*

In terms of literature citations, the 626 papers generated a total of 29,284 citations. The most cited paper with 1066 citations was <Consumption-based accounting of $CO_2$ emissions> by Davis SJ and Caldeira K [5]. There are 71 papers cited more than 100 times, reflecting the research enthusiasm. Table 1 shows the top 10 cited papers. The corresponding authors of these papers are all from developed countries, showing the academic heritage of the old powerhouse.

**Table 1.** Highly cited papers on embodied carbon.

| Author | Times of Cite | Country | Publish Year | Journal |
|---|---|---|---|---|
| Davis SJ and Caldeira K [5] | 1066 | USA | 2010 | Proceedings of the National Academy of Sciences of the United States of America |
| Peters GP and Minx JC et al. [6] | 904 | Norway | 2011 | Proceedings of the National Academy of Sciences of the United States of America |
| Wiedmann, T [7] | 596 | UK | 2009 | Ecological Economics |
| Wiedmann T and Lenzen M et al. [8] | 588 | UK | 2007 | Ecological Economics |
| Lenzen M and Kanemoto K et al. [9] | 578 | Australia | 2012 | Environmental Science & Technology |
| Feng KS and Davis SJ et al. [10] | 446 | USA | 2013 | Proceedings of the National Academy of Sciences of the United States of America |
| Mi ZF and Zhang YK et al. [11] | 381 | UK | 2016 | Applied Energy |
| Meyfroidt P and Rudel TK et al. [12] | 356 | Belgium | 2010 | Proceedings of the National Academy of Sciences of the United States of America |
| Lenzen M [13] | 332 | Australia | 1998 | Energy Policy |
| Weber CL and Matthews HS [14] | 326 | USA | 2008 | Ecological Economics |

*2.3. Region Statistics*

Geographically, China is the most popular country to be studied, involving 425 papers which account for 68% of the total number of papers. Especially after 2010, the number has been spurting, which is closely related to the rapid development of China's economy, the year-on-year growth of trade volume since its accession to WTO, and the increase of the Chinese government's efforts on scientific research (a total of 307 publications were funded by the National Natural Science Foundation of China (NSFC)). Figure 2 lists the top ten hot

areas and the percentage of all studies they account for. Except for China, other countries in the top 10 are all trading powerhouses, reflecting the impact of trade on carbon emission research.

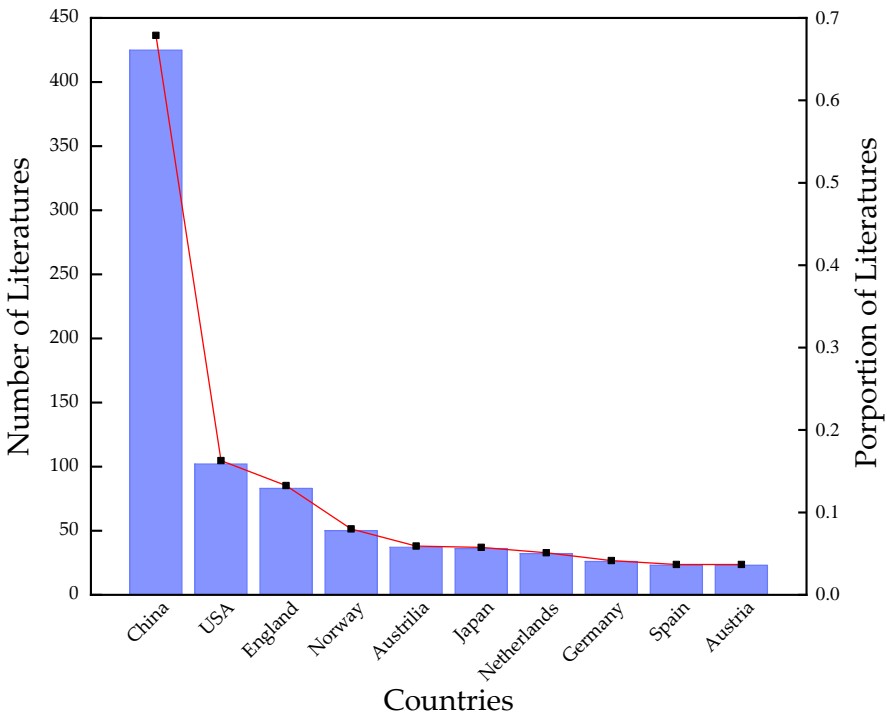

**Figure 2.** Research region on embodied Carbon.

### 2.4. Author Statistics

From the perspective of research scholars, Chen B, a scholar from Peking University, published 29 papers, which is the most prolific of an author in this field, and the average citations of his papers reached 49, showing the quality of these achievements. Table 2 lists the top ten authors with the number of publications, of which seven are from China, which is an overwhelming number. Another two scholars are from the UK, and one is from Norway. Although Chinese scholars do not have an advantage in terms of the highest number of citations, the trend to improve the number of citations in the future is quite evident with the overall latecomer advantage and the trade increase of China in recent years.

**Table 2.** Prolific authors on embodied carbon.

| Name | Number of Papers | Cite Number | Per Cite | Country |
|------|------------------|-------------|----------|---------|
| Chen B | 29 | 1425 | 49 | China |
| Chen GQ | 28 | 2455 | 88 | China |
| Guan DB | 27 | 2667 | 99 | UK |
| Meng J | 27 | 1494 | 55 | UK |
| Zhang B | 22 | 825 | 38 | China |
| Wang Q | 21 | 423 | 20 | China |
| Peters GP | 20 | 3321 | 166 | Norway |
| Liu Y | 19 | 453 | 24 | China |
| Zhang ZK | 14 | 669 | 48 | China |
| Feng KS | 13 | 939 | 72 | China |

### 2.5. Journal Statistics

About publications, the journal with the highest number of publications is the Journal of Cleaner Production, which accumulated 72 papers, and its impact factor is 11.072 (2021), showing the importance of embodied carbon. It is followed by Energy Policy and Ecological Economics, which have 40 and 31 papers, respectively. Both impact factors are above 6. Table 3 lists the top 10 journals in terms of the number of papers published, with two from the US, three from the UK, three from the Netherlands, one from Germany, and one from Switzerland. Although the total number of journals studying China accounts for 68%, there are no journals from China in the top ten.

**Table 3.** Journals of published papers on embodied carbon.

| Name | Number | Indicator (2021) | Country |
|------|--------|-------------------|---------|
| Journal of Cleaner Production | 72 | 11.072 | USA |
| Energy Policy | 40 | 7.576 | UK |
| Econological Economics | 31 | 6.536 | Netherlands |
| Applied Energy | 30 | 11.446 | UK |
| Environmental Science and Pollution Research | 29 | 5.19 | Germany |
| Energy Economics | 27 | 9.252 | Netherlands |
| Sustainability | 25 | 3.889 | Switzerland |
| Science of the Total Environment | 24 | 10.753 | Netherlands |
| Environmental Science & Technology | 20 | 11.357 | USA |
| Journal of Environmental Management | 17 | 8.91 | UK |

### 2.6. Discipline Statistics

In terms of the research discipline, the research of embodied carbon is a relatively complex interdisciplinary study involving 31 research categories. The highest five correlations are Environmental Sciences Ecology, Science Technology Other Topics, Business Economics, and Engineering and Energy Fuels, as shown in Table 4. The share of Environmental Science Ecology alone is nearly 70%, indicating that embodied carbon and carbon emissions are still researched based on environmental sciences. The intersection with various disciplines shows the complexity of this study and the increasing demand for background knowledge in environmental science, economics, mathematics, physics, etc.

**Table 4.** Research disciplines on embodied carbon.

| Discipline | Number | Percent |
|------------|--------|---------|
| Environmental Sciences Ecology | 427 | 0.68 |
| Science Technology Other Topics | 160 | 0.26 |
| Business Economics | 156 | 0.25 |
| Engineering | 148 | 0.24 |
| Energy Fuels | 116 | 0.19 |

## 3. Research Methodology

The mainstream quantitative methods of trade-embodied carbon research are mainly based on Input–Output Analysis (IOA). IOA was first proposed by Leontief in the 1930s, that is, to build a mathematical model and analyze the relationship between supply and demand based on the input and output tables of a particular sector within a certain period. After years of development, it has gradually evolved into Single Region Input–Output

(SRIO), Bilateral Trade Input–Output (BTIO), Multi-Regional Input–Output (MRIO), etc. Table 5 briefly describes the application and advantages of the three methods.

**Table 5.** Advantages of the three IOA methods.

| Method | Research Application | Advantage |
|---|---|---|
| SRIO | Study a single region or a country. | The amount of data required is small and the operation is simple and easy. |
| BTIO | Study bilateral trade between two countries or regions. | The results are more reasonable when the carbon emission factors of the two countries are considered separately. |
| MRIO | Comprehensively analyze the embodied carbon level of a country or region. | Full consideration of intermediate goods in exports and imports and carbon emission factors of each of the trading partners. |

*3.1. Single-Region Input–Output Model*

The Single-Region Input–Output model is mainly applied to the case of a single country or region in the study of embodied carbon in international trade. By using a country's input–output table and data, the embodied carbon of the country is measured under the assumption that other countries have the same carbon emission factors and no carbon emission differences from the country. Using a single-region input–output model, Schaffer R and deSA AL examined the energy and carbon emissions contained in Brazil's non-energy exports and imports during 1970–1993. They found that the amount of embodied carbon exported by Brazil since 1980 was much higher than that imported, and the total embodied carbon exported by Brazil in 1990 reached 11.4% of the national carbon emissions, which was nearly 8.3 million tonnes [15]. Scachez-Choliz J and Duarte R analyzed the relationship between international trade and air pollution in Spain by the IO model and found that Spain conveys a slightly exporting behavior in the Spanish economy, with exports concentrated in transport materials, energy, mining, and chemical industries [16]. Kondo Y and Moriguchi Y et al. used input–output tables to estimate the amount of carbon dioxide contained in Japanese imports and exports and observe the trend. Japan was a net embodied carbon exporter until 1985 when it was assumed that imports and exports had the same carbon intensity. However, the situation started to change after the government implemented the policy of expanding domestic demand in 1985, and it became an embodied carbon importer after 1990 [17]. Weber CL and Peters GP et al. analyzed China's $CO_2$ emissions and found that the share of $CO_2$ emissions from China's exports increased from 12% (230 Mt) in 1987 to 21% (760 Mt) in 2002 and that about 1/3 of China's carbon emissions (1700 Mt) in 2005 came from exports. Exports contributed to China becoming the world's largest emitter of $CO_2$ [18]. Peters GP and Hertwich EG calculated carbon dioxide emissions from 87 countries in international trade in 2001 through input–output analysis. They argued that carbon emissions in trade can have a significant impact on the participation and effectiveness of global climate policies such as the Kyoto Protocol, and countries should pay more attention to the binding and integrity of the agreement when formulating environmental policies [19]. Guo S and Li YL et al. used a multi-scale single-region input–output model to analyze the energy use patterns of Beijing and Shanghai, showing that the energy consumption of the two megacities comes almost entirely from domestic and foreign imports, and the growth of cities has put enormous energy pressure on surrounding areas and led to their environmental degradation [20].

The single-region input–output model is simple to operate and requires less data. Due to its assumption of the same carbon emission factors, it ignores the influence of different energy structures and production levels in each country, which will inevitably produce some errors in the calculations.

### 3.2. Bilateral Input–Output Model

The bilateral input–output model is similar in principle to the single-region input–output model. It considers the difference in carbon emission coefficients between two countries in a bilateral trade situation and can be used to measure the embodied carbon more reasonably according to the situation of each trading country. Thus, it is more favored by scholars in the study of bilateral trade. Shui B and Harriss RC examined the impacts of US–China trade on $CO_2$ emissions in both countries and the world and found that US–China trade increased global $CO_2$ emissions by 7.2 million metric tons between 1997 and 2003. About 7–14% of China's $CO_2$ emissions are caused by producing exports for US consumers. If goods were produced in the US, it would increase US $CO_2$ emissions from 3% to 6% [21]. Li Y and Hewitt CN calculated $CO_2$ emissions from China–UK trade and demonstrated that through trade with China, the UK reduced $CO_2$ emissions by 11% in 2004. Due to the greater intensity of China's carbon emissions and the inefficiency of its production process, China–UK trade led to an increase in global $CO_2$ emissions of 117 Mt in the same year, which was equivalent to 14% of the UK's annual carbon emissions and 0.4% of global emissions in that year [22]. Zhao YH and Wang S et al. used a bilateral input–output model to analyze $CO_2$ data from China–Japan trade between 1995 to 2009 and found that China's $CO_2$ emissions from exports to Japan increased by nearly 100% over this period, and imports also increased by more than 500%. Overall, China is still a net exporter, and emissions increased from 96.15 Mt to 161.59 Mt, mainly reflected in the heavy industry trade [23]. Ackerman F and Ishikawa M et al. studied the carbon emissions between Japan and the US. They found that the trade between Japan and America reduced global carbon emissions, and America transferred part of its $CO_2$ emissions to Japan. Although Japan is an export country of carbon emissions in this trade, its export ratio is very low. As large importers of embodied carbon, their environmental burden is mainly borne by other countries [24].

The bilateral input–output model is more accurate and reasonable for calculating trade between two countries, but it is only an extension of the single-region input-output model. With the gradual development of global trade and the emergence of various regional groups, this model is slightly less capable of dealing with the multilateral nature and complexity of trade.

### 3.3. Multi-Regional Input–Output Model

Multi-regional input–output model is applied in more complex embodied carbon studies. Compared with the single-region input-output model and bilateral input–output model, it considers final and intermediate consumer goods in trade, emphasizes the carbon emissions in processing trade, and considers that different countries have different levels of carbon emission. Due to its wide coverage and comprehensive consideration of various factors, this model has become the most popular model among scholars. Of the 626 papers counted in this study, more than half of them used the MRIO to conduct this research. Oppon E and Acquaye A et al. studied bilateral trade between the UK and 27 African countries. They found that African countries are at a trade disadvantage in the use of carbon, land, and water and that such carbon leakage is mainly concentrated in energy-intensive sectors such as electricity, gas, and oil [25]. Fan JL and Dong YY et al. used an improved MRIO model to analyze the impact of the "Belt and Road" policy and trade diversion on China's $CO_2$ emissions in the context of Sino–US trade friction. The results showed that both import and export trade shifts increase global trade $CO_2$ emissions by 81.76 Mt, which is mainly due to the export of $CO_2$ from Russia, India, and Southeast Asia to China [26]. Ji X and Liu YF et al. studied China–Africa trade by MRIO and demonstrated that China is in a carbon export position in this trade. During 2000–2012, the embodied carbon emissions of bilateral trade gradually increased, but steadily decreased after 2012, and both sides achieved some success in energy conservation and emission reduction [27]. Wang TR and Chen Y et al. analyzed the trade carbon emissions among 31 provinces in China through the MRIO model. They argued that the total scale of embodied carbon

emissions continued to rise from 2012–2017 due to the rapid increase in the economy, and the share of carbon emissions that increased from 13.66% to 20.39% gradually tended to be concentrated in a few provinces [28]. Yuan R and Rodrigues JFD et al. examined the impact of US–China trade on global $CO_2$ emissions in the context of the current US–China trade war. They find that if trade between two countries is completely decoupled, the shift of demand and supply markets for commodities will result in a net increase in global carbon emissions of 0.3–1.8%. In contrast, the supply of trade with the US would shift mainly to other Asian countries, resulting in a 1.2–5.7% increase in their carbon emissions [29].

In addition to international trade, MRIO has a wide range of applications in the study of inter-regional carbon footprint and carbon emissions. Kanemoto K and Shigetomi Y et al. used MRIO to construct carbon footprint inventories for 1172 Japanese cities and found that 40% of Japan's total carbon footprint was driven by 143 cities and that the more densely populated and higher income cities had higher per capita urban carbon footprint emits [30]. Liu GY and Casazza M et al. studied the environmental impact of the coal industry supply chain and coal consumption in the Beijing-Tianjin-Hebei region of China. They said that the proportion of direct use of raw coal is high, and coke is the main source of energy for final consumption. If the government wants to reduce energy consumption and carbon emissions, controlling industrial use and improving processes is the only option to choose [31]. Ivanova D and Vita G et al. developed an inventory of carbon footprints associated with household consumption for 177 regions in 27 countries, showing that GHG emissions are mainly driven by socioeconomic factors (income, household size, education, housing, etc.) and less by geography and infrastructure [32].

Although the MRIO model is widely popular among scholars, it is more complex to operate and has a greater demand for data. Considering that a large number of assumptions are often required in the modelling process, and that there is a more obvious lag in updating the multi-regional input–output database, the accuracy of this model in use is relatively low.

### 3.4. Other Methods

Besides input–output models, Life Cycle Assessment (LCA), Logarithmic Mean Divisa Index (LMDI), and General Equilibrium models are also commonly used in embodied carbon studies, while Index Decomposition Analysis (IDA) [33,34] and Structural Decomposition Analysis (SDA) [35,36] are often used as data processing tools along with the IOA to calculate the embodied carbon. Sevigne-Itoiz E and Gasol CM et al. used Consequential Life Cycle Assessment (CLCA) to evaluate the environmental performance of waste aluminum recycling in Spain and GHG emissions due to market changes in the aluminum industry, showing that increasing the export of aluminum waste can reduce GHG emissions by up to 250% compared with recycling old waste locally [37]. Wang Q and Wang SS studied the changes in carbon emissions and their influencing factors after the COVID-19 pandemic by LMDI. Using examples of global carbon emission changes after the 2008 global financial crisis, they found that middle- and high-income groups are the largest drivers of carbon emission increases while expanding trade openness and improving energy use efficiency would curb the increase [38]. Springmann M and Zhang D et al. used the Computable-general-equilibrium model to analyze the economic impact of China's 12[th] Five-Year Plan on energy conservation and emission reduction in each province. The results showed that 14% of carbon emissions from the eastern provinces are transferred to the central and western provinces. Adjusting the emission reduction plan targets would increase the burden of reducing emissions by 60% in eastern provinces and reduce it by 50% in central and western provinces while doubling China's national welfare losses [39]. Hotak S and Islam M et al. assessed the long-term correlation between carbon emissions and carbon trade balance by a panel-pooled mean group-autoregressive distributive lag model (PMG-ARDL). They said that there is a correlation between carbon emissions and carbon balance trade in high-income countries, while this is not significant in low-income countries. High-income countries achieve carbon emission reductions by outsourcing high-intensity production units and trade carbon replacement [40]. In addition, Vector Autoregression (VAR) [41]

and the Dynamic Integrated Model of Climate and Economy (DICE) [42] have also been used by scholars. Others have discussed this from the perspective of environmental policy. McGee J and Taplin R discussed the potential impact of the APP agreement (2005 Asia Pacific Partnership on Clean Development and Climate) on the willingness of developed countries to save energy and reduce emissions [43]. In addition, Rodrigo MNN and Perera S et al. proposed a novel method for calculating embodied carbon-SCEEM (the Supply Chain based Embodied carbon Estimating Method) [44]. This method, mainly applied to the construction industry, integrates the concepts of the supply chain, value chain, and blockchain to assign values to each stakeholder in the production process to facilitate an accurate embodied carbon estimation.

In summary, input–output models have been widely and maturely applied in the field of international trade-embodied carbon research. With the maturity of research methods and model revision, the accuracy of input–output model estimation is also gradually improved. Considering the increasingly serious environmental problems, the intersection and integrated research of environmental science, economics, mathematics, and other disciplines will be more frequent in the future, and the selection and use of embodied carbon models will certainly be more reasonable and efficient.

## 4. Embodied Carbon Impact Factors

### 4.1. The Impact of International Trade on Carbon Emissions

Since the establishment of the WTO organization in 1995, global trade has surged and the total trade reached a record high of $28.5 trillion in 2021 [45]. The prosperity of trade inevitably comes at the cost of the depletion of natural resources and the environment, and the developed trade exchange system makes it easier to transfer carbon emissions. A country is able to import goods or energy to reduce carbon emissions needed to produce essential goods in its own country, thus transferring pollutant emissions to the country where the goods are produced, resulting in an embodied carbon transfer. This is particularly evident in developed countries. Wyckoff and Roop were the first scholars to calculate data on embodied carbon in international trade. Using OECD data on trade flows, input and output, and energy use, they estimated that the six largest OECD countries (America, the UK, Japan, Germany, France, and Canada) account for 13% of their total embodied carbon in the manufactured goods they import [4]. Ahamad and Wyckoff's results showed that OECD countries consumed 5% more $CO_2$ than they produced. This gap was mainly caused by the US, Japan, France, Germany, and Italy, and the US alone accounted for half of the gap. China and Russia are net exporters of carbon emissions, producing 10% and 15% more than their consumption [46]. Through the observation of 113 countries and regions around the world, Peters and Minx et al. constructed an annual time-series from 1990 to 2008 of $CO_2$ emission inventories based on consumption by adjusting territorial emission inventories with the estimates of net emission transfers via international trade. They calculated that carbon emissions from the production of goods and services have risen from 4.3 Gt $CO_2$ (20% of global emissions) in 1990 to 7.8 Gt (26%) in 2008, and the transfer of emissions from developing countries to developed countries through international trade increased from 0.4 Gt $CO_2$ in 1990 to 1.6 Gt $CO_2$ in 2008 [6]. Jiang XM and Guan DB explored the driving forces of $CO_2$ emissions in OECD and non-OECD countries from 2008 to 2011, showing that the increase in consumption and investment in non-OECD countries was the main reason for the growth of global carbon emissions after 2009. Both OECD and non-OECD countries began to increase purchases of intermediate and final products from non-OECD countries to control their own $CO_2$ emissions [47].

In addition to simply calculating $CO_2$ emissions, many scholars have also used mathematical tools to analyze the relationship between embodied carbon and economic indicators. Essandoh OK and Islam M et al. used the PMG-ARDL model to analyze the relationship between $CO_2$ emissions, international trade, and FDI in 52 countries from the perspective of investment and found that every 1% increase in FDI in developing countries would result in a 0.25% increase in $CO_2$ emissions [48]. Leitao NC and Lorente DB evaluated

the relationship between economic growth, trade openness, and $CO_2$ emissions in the EU region by using FMOLS, DOLS, and the SMM-System, and verified that carbon emissions are positively related to economic growth and negatively related to trade openness [49]. Rahman MM and Alam K et al. used Driscoll and Kraay's standard error and panel-corrected standard error (PCSE) model to observe the effect of the studied variables on $CO_2$ emissions. Combined with the EKC hypothesis, they confirmed a bidirectional causal relationship between international trade and $CO_2$ emissions. That is, international trade damages the environment by increasing $CO_2$ emissions but the square of economic growth reduces its impact [50], as confirmed by the research of Kang H [51]. The study by Du KR and Yu Y et al. examined the effect of international trade on the performance of carbon dioxide emissions from the income perspective. When a country's per capita income level is below \$16,883.45, international trade hinders $CO_2$ emission performance; when income lies between \$18,833.45 and \$33,766.9, there is no significant effect; and when income is greater than \$33,766.9, international trade is beneficial in improving $CO_2$ emission performance [52]. This is consistent with the aforementioned tendency of developed countries to reduce their own emissions by purchasing commodities.

Considering the geographical nature and complexity of international trade, many scholars have chosen a single country as the main subject of study to examine the carbon emission performance of different countries in trade. Weber CL and Matthews HS analyzed data from 1997–2004 for the US and its seven major trading countries and found that the US imports of $CO_2$ rose from 0.5–0.8 Gt in 1997 to 0.8–1.8 Gt in 2004, accounting for 20% of national emissions [53]. Mutascu M explored the correlation between trade openness and $CO_2$ emissions in France. He argued that there is no significant correlation between the two in the short term, and trade openness stimulated $CO_2$ emissions in the long run. While emissions can be controlled by environmental and trade policies [54]. Machado G and Schaeffer R et al. evaluated international trade on Brazil's energy use and $CO_2$ emissions and found that Brazil's non-energy exports contained 12.5 Mt and imports of 9.9 Mt of embodied carbon in 1995, while Brazil's total carbon emissions in that year were 99.4 Mt, with an export share ratio of nearly 14% [55]. Sun CW and Chen LY et al. confirmed the carbon export status of major energy exporters in trade through Russia's carbon emissions in international trade during 1995–2014. A total of 31.46% of its national carbon emissions in 1999 were exported to other countries, although this figure has decreased with the upgrade of traditional manufacturing and the innovation of modern technology, it is still in a disadvantageous position in trade [56].

From the above studies, although the data, models, and research methods used in different studies are inconsistent and the selected research objects are not the same, the basic conclusions are relatively consistent. That is, international trade plays a significant role in global carbon emissions and greenhouse gas control, and developed countries have realized the transfer of carbon emissions through the form of international trade, which effectively reduces the pressure of carbon emissions in their own countries. Therefore, in future international negotiations, it should be clarified that greenhouse gas emissions should not be limited to individual countries, but ought to confirm their respective environmental responsibilities based on their actual production and consumption, as well as pay more attention to the pressure of BRICS countries in carbon emission exports.

### 4.2. International Pollutant Identification Principles

For a long time, the responsibility of international pollutants is mainly determined by the producers, and treaties such as the United Nations Framework Convention on Climate Change (UNFCCC) and the Kyoto Protocol (KP) have implemented producer-based accounting principles. However, this principle only considers the sum of pollutant emissions from all sectors within the producing country, and $CO_2$ used for exports and domestic consumption is included in the producer's carbon emissions account [57]. Under this circumstance, if a country meets its domestic demand for highly polluting products only through import, it can transfer its own carbon emissions to the product-producing

country, thereby reducing its emissions to avoid the identification of pollutant emission responsibility and ensuring its quality of life at the same time. This is contrary to the view of some scholars [43,46]. Therefore, the principle of pure polluter responsibility has been questioned by many scholars. Mozner ZV argued that international trade blurs the eco-efficiency responsibility of producers and consumers, and it would have a longer-term impact on producers. Shifting from producer-based to consumer-based climate policy-making could reduce pollutant emissions [58]. Wang F and Liu BB et al. pointed out that the producer-based accountability system would aggravate the imbalance of regional development [59].

Corresponding to the producer responsibility system is the consumer responsibility principle. This regulation makes up for the shortcomings of the producer responsibility principle, arguing that every economic activity will have an impact on the ecology of the earth, and ecological damage will occupy the public wealth. Therefore, consumers should be responsible for the production process of commodities and for the relevant pollutant emissions. This principle reduced the responsibility of developing countries and raised developed countries' duty, thereby reducing the situation of "carbon leakage". Liddle B believes that high-income countries have the responsibility to help countries such as China and India which are particularly important in the global carbon transfer chain to reduce the intensity of carbon emissions [60]. Pan J and Phillips J et al. compared producer-based and consumer-based carbon emissions in China in 2006 and found that the latter reduced both net emissions and growth rates from 2001 to 2006 [61].

However, the consumer responsibility principle may also raise new issues, such as that producers will be reluctant to actively reduce emissions and to develop and upgrade new technologies to improve energy efficiency. This will make it difficult to achieve the goal of global emission reduction. Thus, many scholars have proposed a joint responsibility principle, which means the two parties are jointly responsible for the discharge of pollutants. Bastianoni S and Pulselli FM et al. proposed an additional method of carbon emissions, which weighs the interests of all parties between consumption and production accounting principles and then allocates them [62]. Csutora M and Mozner ZV also put forward a similar beneficiary-based joint responsibility method, where consumers enjoy the product and producers benefit from the production process, making it so the two share the duty for carbon emissions from their respective benefits [63]. Zheng Z, on the other hand, developed a model to calculate productive carbon emission transfers based on the principle of the top gainer principle and established a traceability mechanism from the perspective of global value chains with a view to avoiding endless debates on producer–consumer accountability [64].

In conclusion, regardless of the consumer responsibility principle or the shared system, it is emphasized that the final consumption or profit is the main cause of environmental pressure. Even the producer responsibility principle's original intention is to reduce global pollutant emissions in order to pursue longer-term benefits. Therefore, how to allocate responsibilities more reasonably based on ensuring energy conservation and emission reduction will be the focus that countries still need to debate at the policy conference.

## 5. Discussion and Conclusions

International trade is a significant factor affecting a country's carbon emissions, and it is also one of the meaningful links in emission reduction. In this study, the Web of Science database of Clarivate is used to count the literature related to international trade embodied carbon and carbon emissions in the past 30 years, and a total of 626 valid papers were obtained. The number of the literature in this research field has gradually increased since 2005, peaking and stabilizing in 2019, as shown in Figure 1. We summarized the obtained literature according to six categories: the number of the literature, citations, research region, authors, journals, and disciplinary classification. In addition, a brief introduction and description of the top volume in each category were provided, and a specific analysis of the existing research methods was presented. Furthermore, we give a brief review of

current research on embodied carbon and carbon emissions in international trade. The main conclusions are as follows:

Firstly, the research on embodied carbon in international trade started in 1994, and the number of the literature has increased significantly since 2010 and exploded in 2018. The underlying reason is that changes in global climate caused by pollutants have become increasingly impossible to ignore, and governments must focus on the issue of sustainable human development. At the same time, third world countries (mainly China) have gradually become major global trading countries, leading to the embodied carbon contained in their exports becoming a research hotspot for scholars in both their own and other countries.

Secondly, input–output models play a prominent role in the study of embodied carbon. Although its performance in measurement accuracy and data requirements is not satisfactory and the results of different scholars for the same country often vary too much, the input–output model will be more widely used in the future as more scholars enter this research field and continuously optimize and improve the model. Moreover, the application of CGE, hybrid MRIO-LCA, LMDI, SCEEM, and other models and methods has greatly enriched the research of embodied carbon.

Thirdly, the current international pollutant identification principles are unfavorable to many industrial countries and third world countries, which only consider the pollutant emissions within national borders and ignore the contribution of upstream and downstream producers and consumers associated with the product to pollutant emissions, resulting in "carbon leakage" from developed countries to developing countries. It makes it more difficult to evaluate the real GHG emissions of each country and is also detrimental to climate policy-making. However, it should be noted that the phenomenon of "carbon leakage" is prevalent in international trade. Just because the trade volume between developed countries and developing countries is considerable and the attention is high, the trade within developed countries and between developing countries cannot be ignored as well.

Since the adoption of the UNFCCC, countries around the world have made full efforts to control pollutant emissions. The EU was the first region to launch a pollutant trading scheme in 2005, and it gradually developed into a carbon trading market. As the world's largest carbon emitter, China also officially opened its own carbon trading market in 2021 and became the world's largest one in the first year of opening. The expansion of carbon markets is vital and indispensable in limiting global pollutants. With more countries announcing that they will achieve carbon neutrality by 2050, the issue of embodied carbon will inevitably become more crucial and will be the focus of discussion among countries in the next round of climate policy conferences. As such, there is an urgent need to clarify how to reasonably calculate the carbon emissions of each country and how to distinguish the responsibilities of consumers and producers in order to achieve a common global pollutant reduction goal as soon as possible.

**Author Contributions:** Conceptualization, H.W. and T.F.; methodology, H.W.; software, H.W.; validation, H.W. and T.F.; formal analysis, H.W.; investigation, H.W. and T.F.; resources, H.W.; data curation, H.W.; writing—original draft preparation, H.W.; writing—review and editing, T.F.; visualization, T.F.; supervision, T.F.; project administration, T.F.; funding acquisition, T.F. All authors have read and agreed to the published version of the manuscript.

**Funding:** This research was supported by the JSPS KAKENHI, Grant Number JP21K01485.

**Institutional Review Board Statement:** Not applicable.

**Informed Consent Statement:** Not applicable.

**Data Availability Statement:** Data are contained within this paper.

**Conflicts of Interest:** The authors declare no conflict of interest.

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
