# Peer review of "A Review of Research on Embodied Carbon in International Trade"

_sustainability, doi:10.3390/su15107879_

Round 1
Reviewer 1 Report
Thanks for the opportunity to review this article. I appreciate the effort the authors put into this research. The analysis carried out by the authors is an interesting one, representative of the literature. The article is well structured, well written, the analysis tools used are appropriate, the discussions are pertinent. However, I recommend writing a chapter of conclusions, writing bibliographic sources for tables and figures and correlating them with the text of the article. I also recommend specifying the software used in the analysis and the databases used in the bibliography.
Author Response
Thank you very much for your review comments.
Regarding your two recommendations, I have revised the first paragraph of the conclusion section of my paper, adding a brief summary of the database and graphs used, and explaining the source of the literatures. Please let me know if you have any other suggestions.
Thank you.
Reviewer 2 Report
Intresting paper analyzing current available literature regarding GHG emission reduction. Introduction: Can you add explanation how your data collection and your analyze the development and trends of research conect with (or effect on) importance of GHG emission.
Section 2: can you explain more what you mean vaild here: “ 626 valid papers were obtained..” and how you shoose this paper? Did you take into consideration reports of official organizations? Explain, why focus only on papers?
Fígure 2; whuld be possible compare China with the EU? May be in the EU the number of publications large then in China.
Can you explain more what kind of scholars you used and why. You use chinese scholar (scholar from Peking University) so that why use get large amount of chinese publications. Do you use google scholar, reaserch gate , any others?
Do you make appendix in witch you say what literatures are taken into account in your study.
Can yo give more recomendations for future reaserch? What aspect is missing in current academic litrature?
Author Response
Thank you for your review comments.
Point 1: Introduction: Can you add explanation how your data collection and your analyze the development and trends of research conect with (or effect on) importance of GHG emission.
Response 1: As I can see, GHG emissions encompass several aspects, such as industrial production, transportation, and human daily life. These components, as the most common source of GHG emissions, always receive attention from all parties, but trade embodied carbon is not. Trade embodied carbon is often overlooked due to its specificity, and it can influence countries` mitigation policies (e.g., through the transfer of responsibility for carbon emissions). Considering some international policies such as the Kyoto Protocol and the Paris Agreement, an unclear system of carbon emission recognition and responsibility can lead to mutual conflicts among countries and free-riding, which is very unfavorable for countries to jointly negotiate a global GHG emission reduction policy. In addition, the scale of global trade is growing and the reduction of GHG emissions must go hand in hand. Thus, any area where GHG emissions can be controlled should be given extensive attention.
Point 2: Section 2: can you explain more what you mean vaild here: “626 valid papers were obtained..” and how you shoose this paper? Did you take into consideration reports of official organizations? Explain, why focus only on papers?
Response 2: Valid means useful, or meaningful. I conducted a search of literatures using the Web of Science database owned by Clarivate. In the search results, there were 2 papers that were duplicates and I am not sure if it was due to the database. I eliminated this paper, so the remaining literatures are valid.
I think papers reflect the comprehensive results of academic research and have the highest academic value, so I chose papers as the object of my study. I also considered official organization reports and quote some of them as well (such as IPCC). However, reports mainly show the current situation, while papers try to propose solutions to the existing problems. Therefore, reports are mostly cited in papers as research backgrounds or introductions, hence I chose papers.
Point 3: Fígure 2; whuld be possible compare China with the EU? May be in the EU the number of publications large then in China.
Response 3: I am not comparing the study areas (like China and EU), I just showing the number of papers. It is true that there are probably more papers studying European issues than China (maybe published in each country`s own research platform), but in the Web of Science database, this is the result.
Point 4: Can you explain more what kind of scholars you used and why. You use chinese scholar (scholar from Peking University) so that why use get large amount of chinese publications. Do you use google scholar, reaserch gate , any others?
Response 4: I haven`t use google scholar and any other research tools except Web of Science. All the papers I used were published in SCI or SSCI journals, and I did not use any Chinese publications. I think the reason why Chinese scholars and China-related research appear so frequently I is that China is the world`s top carbon emission emitter and has the largest number of scholars studying Chinese issues.
Point 5: Do you make appendix in witch you say what literatures are taken into account in your study.
Response 5: I did not write an appendix, but all references (paper citations) are from 626 papers I counted, as can be shown in references.
Point 6: Can yo give more recomendations for future reaserch? What aspect is missing in current academic litrature?
Response 6: I hope I can provide ideas on research methods for future studies, just like IOA, LCA, and SCEEM that I talked about in the introduction of methods. I think the study of trade embodied carbon is a hot topic for future research, because trade will keep expanding with the process of globalization, and reducing carbon emissions is a problem that countries have to face. Therefore, when more scholars start to engage in this field, they need to have some understanding of the previous research, whether it is the development history of research background or the continuous updating methods. I hope my research can provide some convenience to future scholars.
I think the current research is still mainly focused on the measurement of embodied carbon, and the division of responsibility is not detailed enough. Considering the general trend of various global climate changes and the comprehensive carbon neutrality plans announced by more and more countries, I think the issue of recognizing responsibility regarding carbon emissions needs to be clarified as soon as possible to prevent free-rider situations, and reduce meaningless bickering.
Thank you very much.
Reviewer 3 Report
Despite not being entirely new, this research is excellent and will be helpful to aspiring scientists. Excellent, well-organized writing can be found throughout the work. The following are some quick suggestions that I'd like to make:
- In the present research, data from the literature on embedded carbon and carbon emissions in global trade during the last three decades have been collected.
- Add the doi and issue no. of all published papers and the report used in the reference section on page no. 13.
- Add 1–2 latest references for the year 2023.
- Modified the boarder title on the x axis as YEARS in line no. 96.
- Add table 1 to the description of section 2.2 citation statistics.
- Add Figure 2 in the description of section 2.2 region statistics and modify the border title on the x axis as countries in line no. 115.
- Table 3, and Table 4 are not available in the discussion of Sections 2.5, and 2.6.
- Add table 5 in the description of section 3. research methodology.
- Suggest authors to add method of calculating embodied carbon given by Wiedmann, T [7]
- Add SCEEM method in research methodology section 3,Line no. 157 from Rodrigo et.al.2021.
Rodrigo, M.N.N.; Perera, S.; Senaratne, S.; Jin, X. Review of Supply Chain Based Embodied Carbon Estimating Method: A Case Study Based Analysis. Sustainability 2021, 13, 9171. https://doi.org/10.3390/su13169171
- Remove the typographical errors.
Author Response
Thank you very much for your comments.
In response to comment 3,4,5,6,7,8, I made revise, please check it.
Regarding comment 2, I added DOI number, but I don`t quite understand the issue no. I checked the reference format requirements and read some new published papers, but I didn`t find issue no. I don`t quite understand how to add it. The same is true for comment 1. If possible, please let me know in detail and I will make changes.
Regarding comment 9, Mr. Widemann`s paper is a review paper, which is a summary of the existing carbon emission research using the MRIO model. The method he introduced and used is consistent with MRIO, and I have introduced and explained this method in the third part of my paper, so I did not add the detailed calculation process of his method.
Regarding comment 10, the SCEEM method you proposed is currently mainly applied in the field of architecture, while my research is mainly based on the environmental economics. This method is rarely used in international trade research, so I put it in the last part of method introduction (other methods). I think this is more reasonable, please let me know if you have other comments.
Thank you very much.